# Microtentacle Formation in Ovarian Carcinoma

**DOI:** 10.3390/cancers14030800

**Published:** 2022-02-04

**Authors:** Jocelyn C. Reader, Cong Fan, Eleanor Claire-Higgins Ory, Julia Ju, Rachel Lee, Michele I. Vitolo, Paige Smith, Sulan Wu, Mc Millan Nicol Ching, Emmanuel B. Asiedu, Christopher M. Jewell, Gautam G. Rao, Amy Fulton, Tonya J. Webb, Peixin Yang, Alessandro D. Santin, Huang-Chiao Huang, Stuart S. Martin, Dana M. Roque

**Affiliations:** 1Division of Gynecologic Oncology, Greenebaum Comprehensive Cancer Center, University of Maryland School of Medicine, Baltimore, MD 21201, USA; jreader@som.umaryland.edu (J.C.R.); cf345@cornell.edu (C.F.); smithpaige340@gmail.com (P.S.); mching5@jhmi.edu (M.M.N.C.); grao@som.umaryland.edu (G.G.R.); 2Department of Pharmaceutical Sciences, School of Pharmacy and Health Sciences, University of Maryland Eastern Shore, Princess Anne, MD 21853, USA; 3Department of Physiology, Greenebaum Comprehensive Cancer Center, University of Maryland School of Medicine, Baltimore, MD 21201, USA; eory@som.umaryland.edu (E.C.-H.O.); jju@som.umaryland.edu (J.J.); ralee@som.umaryland.edu (R.L.); 4Department of Pharmacology, Greenebaum Comprehensive Cancer Center, University of Maryland School of Medicine, Baltimore, MD 21201, USA; mvitolo@som.umaryland.edu (M.I.V.); ssmartin@som.umaryland.edu (S.S.M.); 5Department of Chemistry and Biochemistry, Oberlin College, Oberlin, OH 44074, USA; sulan@molecularinstruments.com; 6Division of Biology and Biological Engineering, California Institute of Technology, Pasadena, CA 91125, USA; 7Cellular and Molecular Medicine, Johns Hopkins University School of Medicine, Baltimore, MD 21205, USA; 8Division of Cancer Imaging, Russel H. Morgan Department of Radiology and Radiological Sciences, Johns Hopkins Hospital, Baltimore, MD 21287, USA; 9Department of Microbiology and Immunology, Greenebaum Comprehensive Cancer Center, University of Maryland School of Medicine, Baltimore, MD 21201, USA; easiedu@som.umaryland.edu (E.B.A.); twebb@som.umaryland.edu (T.J.W.); 10Fischell Department of Bioengineering, University of Maryland College Park, College Park, MD 20742, USA; cmjewell@umd.edu (C.M.J.); hchuang@umd.edu (H.-C.H.); 11Baltimore Veterans Administration Medical Center, Baltimore, MD 21201, USA; afulton@som.umaryland.edu; 12Department of Pathology, Greenebaum Comprehensive Cancer Center, University of Maryland School of Medicine, Baltimore, MD 21201, USA; 13Department of Obstetrics, Gynecology & Reproductive Sciences and Biochemistry & Molecular Biology, University of Maryland School of Medicine, Baltimore, MD 21201, USA; pyang@som.umaryland.edu; 14Division of Gynecologic Oncology, Smilow Cancer Center, Yale University, New Haven, CT 06520, USA; alessandro.santin@yale.edu

**Keywords:** ovarian cancer, microtentacle, serous carcinoma, clear cell carcinoma, microtubules, epothilone, ixabepilone, taxane, paclitaxel, intraperitoneal chemotherapy

## Abstract

**Simple Summary:**

Ovarian cancer commonly spreads throughout the peritoneal cavity by exfoliation of malignant cells into ascites. Chemoresistance remains an important therapeutic obstacle. Microtentacles (McTNs) are microtubule-based protrusions that may influence the metastatic potential and chemoresistance profile of free-floating cells. In this study, we define the various microtentacle morphologies that can be observed in detached ovarian cancer cells, and their clustering behavior in relationship to histology, α-tubulin post-translational modifications, β-tubulin isotype, modulators of cortical stiffness, and sensitivity to clinically relevant microtubule-targeting agents. McTNs represent a new therapeutic target for this disease, and an understanding of their biology could have implications for the refinement of therapies, including intraperitoneal drug delivery.

**Abstract:**

Background: The development of chemoresistance to paclitaxel and carboplatin represents a major therapeutic challenge in ovarian cancer, a disease frequently characterized by malignant ascites and extrapelvic metastasis. Microtentacles (McTNs) are tubulin-based projections observed in detached breast cancer cells. In this study, we investigated whether ovarian cancers exhibit McTNs and characterized McTN biology. Methods: We used an established lipid-tethering mechanism to suspend and image individual cancer cells. We queried a panel of immortalized serous (OSC) and clear cell (OCCC) cell lines as well as freshly procured ascites and human ovarian surface epithelium (HOSE). We assessed by Western blot β-tubulin isotype, α-tubulin post-translational modifications and actin regulatory proteins in attached/detached states. We studied clustering in suspended conditions. Effects of treatment with microtubule depolymerizing and stabilizing drugs were described. Results: Among cell lines, up to 30% of cells expressed McTNs. Four McTN morphologies (absent, symmetric-short, symmetric-long, tufted) were observed in immortalized cultures as well as ascites. McTN number/length varied with histology according to metastatic potential. Most OCCC overexpressed class III ß-tubulin. OCCC/OSC cell lines exhibited a trend towards more microtubule-stabilizing post-translational modifications of α-tubulin relative to HOSE. Microtubule depolymerizing drugs decreased the number/length of McTNs, confirming that McTNs are composed of tubulin. Cells that failed to form McTNs demonstrated differential expression of α-tubulin- and actin-regulating proteins relative to cells that form McTNs. Cluster formation is more susceptible to microtubule targeting agents in cells that form McTNs, suggesting a role for McTNs in aggregation. Conclusions: McTNs likely participate in key aspects of ovarian cancer metastasis. McTNs represent a new therapeutic target for this disease that could refine therapies, including intraperitoneal drug delivery.

## 1. Introduction

Worldwide, an estimated 151,900 deaths from ovarian cancer occur annually [1]. Paclitaxel and carboplatin have represented the standard of care for treatment of this disease since 2003 [2], with initial response rates that exceed 70% [3]. Unfortunately, the majority of patients eventually succumb to progressive chemoresistance, making tantamount the elucidation of mechanisms that underlie drug failure.

Paclitaxel is a hydrophobic diterpenoid that alters the equilibrium between soluble tubulin and the polymerized α-/β-tubulin heterodimers that constitute microtubules [4]. Changes in mitosis [5] as well as cellular migration [6] and microtubule trafficking ensue [7]. pPaclitaxel binds β-tubulin and confers microtubule stabilization in part by promoting lateral interactions of protofilaments [8]. Upregulation of the class III isotype of β-tubulin has been associated with paclitaxel resistance through multiple mechanisms, including reduced binding affinity of paclitaxel. Epothilones, such as ixabepilone, share an overlapping but non-identical binding site [9] that may allow this class of microtubule-stabilizing agents to overcome drug resistance conferred by the effect of tubulin mutants on the binding pocket. Relative to paclitaxel, epothilones also resist cellular export by p-glycoprotein pumps [10]. We have shown previously that ixabepilone is a highly effective agent in heavily pre-treated resistant/refractory ovarian cancers [11,12].

Microtentacles (McTNs) are microtubule-based protrusions supported by vimentin that were first observed in detached breast cancer cells [13]. In breast cancer, they contribute to re-attachment as well as homotypic and heterotypic aggregation [14], and thus have the potential to influence invasion and metastasis. Their growth is regulated by kinesins [15] and microtubule-associated proteins such as tau [16]. Formation is tempered by characteristics of the actin cortex of the cell. Paclitaxel appears to promote McTN extension in adherent but not floating cells, as well as the attachment of floating breast cancer cells to the extracellular matrix [17].

At diagnosis, 80% of ovarian cancers exhibit intra-abdominal metastases above the pelvic brim and at least one-third will exhibit malignant ascites [18]. Given that this environment is rich in floating cells, we hypothesized that microtentacles could represent a key element of ovarian cancer pathophysiology and an important target for therapeutic intervention. Herein, we provide the first description of McTN production by human ovarian cancer cells using a lipid-tethering technique [19] that allows for single-cell analyses in a free-floating state. We characterize ovarian cancer McTNs according to histology, α-tubulin post-translational modifications, β-tubulin isotype, sensitivity to clinically relevant microtubule-targeting agents, modulators of cortical stiffness, and ability to cluster. 

## 2. Materials and Methods

### 2.1. Immortalized Cell Cultures, Tethering and Imaging

Seven serous (OSC) (Kuramochi, OVSAHO, CaOv3, COV-362, SKOV3, OVCAR3, OV90), eight clear cell (OCCC) (CC-ARK1/2, OVTOKO, OVISE, JHOC-5/7, TOV21G, OVMANA), and one clear cell-like (ES2) [20] ovarian cancer cell lines were available for analysis. Human/immortalized ovarian surface epithelium (HOSE/IOSE) was utilized as a control. All cell lines were purchased commercially except CC-ARK 1 and CC-ARK 2, which were donated by Alessandro Santin, MD, (Yale University). Attached cells, as indicated, were cultured at 37 °C at 5% CO_2_ in phenol-free RPMI 1640 (Kuramochi, CC-ARK-1, CC-ARK-2, OVTOKO, OVSAHO, OVISE, OVMANA) or DMEM (4.5G/L) (CaOv3, SKOV3, OVCAR3, COV-362) or 1:1 DMEM: HamF12 (JHOC-5/7) with 10% fetal bovine serum and 1% penicillin/streptomycin. HOSE/ IOSE, OV-90, TOV-21-G cells were grown in 1:1 Medium 199: MCDB 105 with 15% FBS and 1% penicillin/streptomycin. At 70–80% confluence, plates were washed with 2× Dulbecco’s phosphate-buffered saline (DPBS) and then incubated with CellMask (Life Technologies, Carlsbad, CA, USA) at 37 °C for 15 min, washed, trypsinized, and counted with trypan blue. A microfluidic slide (ibidi #80601) coated with five cytophobic polyelectrolyte multilayers (PEMs) was incubated for 30 min with 12,000–24,000 cells in 60 µL as described previously [21]. Cells were imaged with confocal microscopy at ambient room temperature.

### 2.2. Primary Cell Lines and Patient-Derived Samples 

Ascites was harvested from a patient with recurrent platinum- and taxane-resistant stage IV high-grade serous fallopian tube carcinoma under protocol GCC1488, approved by the Institutional Review Board of the University of Maryland-Baltimore. Informed consent was obtained, and the study was conducted in accordance with the Declaration of Helsinki. Ovarian cancer cells were isolated immediately by centrifugation for examination and then cultured for serial passage. 

### 2.3. Chemosensitivity

IC_50_ (half maximal inhibitory concentration) was calculated using Prism 7.0 (GraphPad, San Diego, CA, USA) through interpolation of sigmoidal curves fit with a standard Hill slope of −1.0 after treating cell lines with paclitaxel and ixabepilone along a dynamic range of 0.1–1000 nM for 72 h. Cell viability was assessed using a fluorescence-based microplate reader (CyQuant, Thermo-Fisher, Waltham, MA, USA) according to the manufacturer’s instructions. To examine the effect of drugs on McTNs, cell lines were treated with appropriate vehicle control or drug for 15 min and then tethered and imaged as described above. Drugs were purchased commercially.

### 2.4. Clustering Assays

Cells were allowed to reach 80% confluence in a 10 cm dish. Serum-containing medium was removed. Cells were washed with DPBS and incubated with CellMask at 1:2000 for 15 min, washed with PBS, trypsinized, and counted. Cells at 125,000 cell/well were treated with drugs and allowed to cluster for 18–20 h. Imaging was performed on a Nikon fluorescent microscope in three phases: an initial image, and two more sets of images after agitating the cells on a plate reader for 60 s each for a total of three sets of images.

### 2.5. Western Blot Analyses

Total protein lysates were generated using RIPA buffer (Sigma Aldrich, St. Louis, MO, USA) according to manufacturer’s instructions, supplemented with protease inhibitors, and separated on 10% TGX gel (Bio-Rad, Hercules, CA, USA) under reducing conditions. The gel was transferred to a PVDF membrane and probed with 1:3000 class III β-tubulin (TUJ1; BioLegend, San Diego, CA, USA), 1:10,000 GAPDH (Cell Signaling Technologies, Danvers, MA, USA), Cofilin (D3F9) (1:1000, Cell Signaling Technologies), p-Cofilin (Ser3) (1:1000, Cell Signaling Technologies), monoclonal α-tubulin (DM1A) (1:2000, Sigma Aldrich), anti-detyrosinated α-tubulin (glu-tubulin) (1:1000, Abcam, Waltham, MA, USA), tyrosinated α-tubulin (1:2000, Sigma Aldrich), acetyl α-tubulin (Lys40, D20G3) (1:1000, Cell Signaling Technologies), α-Actin (AC-15) (1:10,000, Sigma Aldrich, Burlington, MA, USA), CAPZB (1:1000, Bio-Rad), and Diap1 (1:1000, Cell Signaling Technologies). Standard molecular weight markers were used (Precious Plus Protein Standards #1610374/ #1610375, BioRad, St. Louis, MO, USA). Densitometry was analyzed with ImageJ: https://imagej.nih.gov/ij/ (accessed on 11 June 2018). Experiments were performed in duplicate and representative images are shown. Raw western blot figures can be found in the Appendix A. 

### 2.6. Microscopy

Confocal microscopy. Images were collected as previously reported [21]. Briefly, imaging was conducted on an Olympus IX81 microscope with Fluoview FV-1000 confocal laser scanning system at 60× magnification. Tethered cell videos were imaged at 1 µm/slice z-stacks every 10 seconds.

Nikon microscopy. Images for the clustering assays were collected on a Nikon Exlipse Ti2-E inverted microscope at 4× magnification on the TRITC channel. Whole channel stitched images were acquired using High Content Analysis.

### 2.7. Analyses

The slide construction, tethering protocol, MATLAB techniques for determination of McTN phenotype, and temporal correlation coefficient were published previously [19,21]. Cluster assays were analyzed by counting the number of segmented objects within each image, which were found using a Laplacian of Gaussian filter. The code used for this segmentation is available at https://github.com/ScientistRachel/CellAggregationAnalysis (accessed on 01 July 2019). Statistical analysis of McTN length; number; and length vs. number for HOSE/IOSE, OCCC, and OSE were performed using a one-way ANOVA analysis with a Tukey Post Hoc analysis using Graphpad Prism 9. Statistical analyses as indicated were performed using Graphpad Prism 9. 

## 3. Results

### 3.1. Immortalized Ovarian Cancer Cell Lines Display Several McTN Morphologies

We observed by confocal microscopy four crude patterns of McTN expression in established ovarian cancer cell lines: absent (A), symmetric-short (SS), symmetric-long (SL), and tufted (T) (Figure 1a). Cells were characterized as ‘absent’ phenotype if they were void of McTNs, ‘tufted’ if they displayed McTN polarized to one area of the cell membrane, ‘symmetric short’ if they demonstrated McTNs circumferentially that did not exceed 1/3 of the cell diameter, and ‘symmetric long’ if they demonstrated McTNs circumferentially with at least one McTN ≥1/3 of the cell diameter. Multiple morphologies were often observed within the same cell line. OSC cell lines displayed a higher frequency of SL morphology, while OCCC cell lines displayed mostly SS or A phenotype. Only OCCC JHOC-7 failed to produce any McTNs (Table 1). Additional characteristics of the patients (e.g., age, stage, treatment status) from whom the cell lines were derived are included in Appendix A. 

### 3.2. Nearly One-Third of All Ovarian Cancer Cells May Exhibit McTNs 

Fluorescence microscopy revealed that 0–29% of cells from any given cell line exhibited McTNs (Figure 1b). The median percent of cells displaying McTNs for OCCC and OSC cell lines was 9.6% (mean: 11.8 ± 3.1%) and 20.8% (mean 17.9 ± 3.6%), respectively (*p* = NS (notsignificant), 0.14).

### 3.3. Freshly Procured Ovarian Cancer Cells from Ascites Also Demonstrate McTNs

After observing the formation of McTNs by immortalized cell lines, we sought to identify whether McTNs might similarly be observed on OSC cells from freshly procured ascites (OV47). Non-adherent cancer cells were isolated by centrifugation, stained, and prepared for tethering using the same protocol employed for cell culture. We observed robust McTN formation in cancer cells obtained directly from ascites (Figure 1c), which persisted through at least two passages of primary cell culture. 

### 3.4. OCCC and OSC Cell Lines Demonstrate Differences in Average McTN Length and Number Per Cell, Suggesting That McTN Morphology Varies by Ovarian Cancer Subtype

OCCC expressed fewer McTNs than OSC (5.72 ± 1.56 vs. 12.42 ± 1.77, *p* = 0.028). OCCC McTNs were also shorter (1.00 ± 0.09 vs. 1.43 ± 0.14 μm, *p* = 0.032). In order provide an objective automated aggregate assessment of McTN morphology, a MATLAB-based algorithm was previously developed [21] to measure the product of length and number of McTNs per cell (Figure 2 inset). The length multiplied by the number of McTNs per cell was lower in OCCC compared to OSC cells (7.88 ± 2.6 vs. 22.00 ± 4.63, *p* = 0.025) (Figure 2). OSC cell line McTNs were longer than those on HOSE/IOSE cell lines (1.43 ± 0.14 μm versus 0.85 ± 0.15 μm, *p* = 0.037). There were no significant differences between length, number, and length multiplied by number for OCCC versus HOSE/IOSE cell lines.

### 3.5. OCCC Overexpress Class III ß-Tubulin (TUBB3) Relative to OSC, and OSC/OCCC Cell Lines Exhibit Microtubule-Stabilizing Post-Translational Modifications of α-Tubulin

Acetylated and detyrosinated tubulin is associated with McTN stabilization and promotion [13,22,23]. Total protein lysates were prepared and analyzed by Western blot and quantified by densitometry for post-translational modifications of α-tubulin (acetylated, detyrosinated, tyrosinated) (Figure 3a,b) or class III ß-tubulin (Figure 3c, serous; Figure 3d, clear cell). We previously showed class III ß-tubulin to be a marker of chemoresistance in OCCC and OSC treated with neoadjuvant chemotherapy [24,25]. Most OCCC exhibited several-fold higher expression of class III ß-tubulin relative to OSC, and some OSC had reduced expression of class III ß-tubulin relative to HOSE. JHOC-7, which does not form McTNs, expressed little acetylated tubulin and class III ß-tubulin.

### 3.6. Microtubule Depolymerizing Drugs Decrease the Number and Length of McTNs, Suggesting That Ovarian Cancer McTNs Are Tubulin-Based

Four OCCC/OSC cells lines were treated with microtubule-depolymerizing (colchicine) and microtubule-stabilizing (paclitaxel, ixabepilone) agents to compare effects on McTN length, number, and length × number. The cell lines exhibited high (TOV21G/SKOV3) and low (OVSAHO/COV362) class III β-tubulin. Colchicine inhibits tubulin polymerization, leading to excess free tubulin that may secondarily limit mitochondrial function [26] and has been shown to destabilize McTNs in mammary cells [21], but its utility is limited by toxicity and narrow therapeutic window [26]. For all cell lines, treatment with colchicine led to a decrease in the average length, number, and length × number of McTNs, re-affirming that these protrusions are tubulin-based (Figure 4). When OSC cell line COV-362 was treated with colchicine, McTN length, number and length × number also decreased. There was no significant difference in any McTN morphological parameter after treatment with paclitaxel or ixabepilone (Figure 4). Cells were treated for 15 min, which should be sufficient to observe effects on McTNs, as paclitaxel can exert an effect on mitotic spindle within as little as 3–5 min [27].

### 3.7. Microtubule-Stabilizing Agents Decrease McTN Dynamics

McTN dynamics can be expressed as cumulative tip distance over time [21] or using a temporal correlation coefficient. This correlation coefficient reflects the similarity between the initial cell image and an image taken at a later time point; thus, a coefficient nearing 1 indicates increasing similarity between the two time points, and therefore fewer fluctuations and more stable McTNs. Two cell lines were chosen based on their differential expression of class III ß-tubulin (OVASHO, an OSC with little/no expression; TOV21G, an OCCC with overexpression) and treated with microtubule-stabilizing agents (paclitaxel and ixabepilone) to evaluate the effect on McTN stability. OVASHO and TOV21G cell lines demonstrated less dynamic McTNs at all three or the last time point, respectively, indicating that treatment with microtubule-stabilizing drugs may also have a stabilizing effect on McTNs (Figure 5).

### 3.8. Cells That Fail to Form McTNs Demonstrate Differential Expression of α-Tubulin Post-Translational Modifications and Actin-Regulating Proteins Relative to Cells That Form McTNs

We next sought to characterize proteins known to modify the actin cortex and influence microtubule assembly using cells in an attached versus detached state (at 30 and 60 min) in relationship to their ability to form McTNs. Actin severing proteins, such as cofilin, promote actin disassembly [28] and generally result in weakening of the actin cortex, favoring the formation of McTNs; phosphorylation of cofilin (p-cofilin) is inactivating [29]. Downregulation of the actin capping protein ‘CAPZB’ and the formin ‘DIAPH1’ also reduces tensile strength of the cellular cortex [30]. The actin isoform β-actin localizes to lamellae, pseudopodia, and cancer cell blebs, and is essential to vascular invasion [31]. α-tubulin detyrosination (i.e., glu-tubulin) can impede microtubule disassembly, promote reattachment, and is associated with metastatic potential and the epithelial to mesenchymal transition [22,32,33]. Consistent with these known functions, the McTN non-forming OCCC cell line JHOC-7 highly expressed p-cofilin in the attached and detached states when compared to OCCC TOV21G, which is associated with SL morphology (Figure 6). CAPZB and DIAPH1 decreased with detachment in TOV21G but increased in JHOC-7. In the attached state, JHOC-7 expressed high levels of tyrosinated α-tubulin, whereas TOV21G favored detyrosinated α-tubulin.

### 3.9. Cluster Formation is More Susceptible to Microtubule Depolymerizing and Stabilizing Agents in Cells that Form McTNs, Suggesting a Role for McTNs in Aggregation

Using standard fluorescence cell viability assays, we then defined the chemosensitivity IC_50_ (mean ± standard deviation) of OCCC versus OSC cell lines to paclitaxel (1.33 ± 1.55 nM vs. 1.74 ± 1.04 nM, *p* = NS) and ixabepilone (1.65 ± 0.82 nM and 2.26 ± 2.22 nM, *p* = NS). IC_50_ values for individual cell lines are detailed in Table 2. We then probed for differences in clustering between cell lines with similar drug sensitivities (i.e., JHOC-7: 1.72 nM and COV362: 2.72 nM) but different McTN morphologies (i.e., JHOC-7: absent, COV362: symmetric long). Cells were incubated for 18–20 h with drug or vehicle and then agitated. Images were taken prior to shaking and after two periods of agitation for 60 seconds each. Clusters within each image were identified and quantified using a custom MATLAB script. COV-362 demonstrated reduced clustering ability (as demonstrated by an increase in the number of smaller clusters) after treatment with a microtubule depolymerizing agent (vinblastine, colchicine), compared to vehicle or stabilizing agent (paclitaxel). JHOC7 demonstrated fewer differences in response to drug treatments (Figure 7), alluding to the participation of McTNs in clustering.

## 4. Discussion

Microtentacles are tapered [34] microtubule-based protrusions void of intermediate filaments that measure approximately 150 nm that have been demonstrated in detached human breast cancer cells [13,35,36] and glioblastoma [37]. McTNs differ distinctly from invadopodia [38] actin-based protrusions that occur on extracellular matrix-based cells that can prompt proteolytic degradation to facilitate invasion, as well as lamellipodia, which are flat, sheet-like structures consisting of long, unbranched actin filaments that govern two-dimensional movement [39]. We provide here the first description of McTNs in human ovarian cancer cells and human ovarian surface epithelium.

In the present study, one-third of all ovarian cancer cells produced McTNs. We deliberately employed two distinct epithelial ovarian cancer histopathologies in this analysis: OSC and OCCC. OCCC demonstrates geographic and ethnic variations; it constitutes 10–25% of all ovarian cancers in Korea, Taiwan, and Japan, but only 1–2% in North America and Europe [40]. OCCC is much more likely than OSC to present at an early stage as a unilateral mass, at a younger age, and in association with endometriosis [41]. While early-stage OCCC is associated with improved progression-free survival, advanced stage OCCC is associated with an overall survival disadvantage (hazard ratio 1.66, 95% CI 1.43 to 1.91) relative to OSC [42]. In accordance with the distinct clinical characteristics and metastatic potential of OCCC, OCCC cell lines displayed a unique McTN morphology (length × number) with McTNs that were shorter and roughly 50% less numerous. The relationship between McTNs and class III ß-tubulin appears to be complex. Future work is needed to address whether McTNs are capable of dysregulated expression of ß-tubulin isotypes, or whether McTNs only incorporate constitutively expressed class I ß-tubulin. Currently, all Food & Drug Administration-approved microtubule-stabilizing agents bind ß-tubulin [43].

Interestingly, normal ovarian epithelium also expressed McTNs. This raises the possibility that some primary peritoneal carcinomas might arise from exfoliated normal ovarian cells that float in physiologic free fluid within the pelvis then adhere to distant sites and undergo malignant transformation in response to a yet unelucidated stimulus. Primary peritoneal carcinoma is predominantly of serous histology but is characterized by widespread intraperitoneal malignancy of the omentum and upper abdomen with minimal or no ovarian involvement [44]. Historically, it has been theorized to arise from malignant degeneration of germ cell rests along the gonadal embryonic pathway or carcinogenesis of coelomic epithelium [45,46]. It is plausible that McTNs in normal ovarian surface epithelium may participate in reattachment for post-ovulatory repair [47] or even neo-oogenesis [48] in those that exhibit or acquire stemness.

While others [37] have successfully used stochastic optical reconstruction microscopy (STORM) [49] to depict McTNs, these investigations were restricted to cells adherent to the matrix. In the present construct, McTNs were imaged in the suspended state, which uniquely permits the study of these structures in a native form. We found that treatment of floating cells with the clinically relevant microtubule-stabilizing agents, paclitaxel and ixabepilone, at doses that greatly exceeded the IC_50_ resulted in stabilization of McTNs in some cells, but was insufficient to alter the formation or length of these structures. This is consistent with previous reports in breast cancer [17]. The reason for this is unknown but overexpression of microtubule-associated proteins such as tau have been shown to promote McTN formation [16] and tau binding may reduce the fraction of open paclitaxel binding sites [50], perhaps negating the effect of paclitaxel on McTNs. Future studies are also required to identify whether McTNs are comprised of class I or class III β-tubulin as overexpression of the latter may also reduce paclitaxel binding [51]. We also observed that treatment with paclitaxel promoted the formation of larger cell conglomerates (i.e., fewer cell clusters). We hypothesize that paclitaxel may confer rigidity to McTNs to allow these aggregates to resist agitation. These findings illustrate the complexities of McTN regulation and prompted the search for other targetable factors that may influence McTN biology.

McTN protrusion is governed in part by the stiffness of the cellular cortex. Cortical tension can be reduced in cells with shorter and thinner actin filaments, which occurs with depletion of DIAPH1 [52]. Actin polymerization also requires actin capping protein (a heterodimer of CAPZA/CAPZB), whose knockdown decreases cortical surface tension [30]. Cofilin is required for actin turnover and fluidization of the actin cortex [30,53]. Consistent with such roles, CAPZB and DIAPH1 decreased upon the detachment of cells that form McTNs, but not in those that fail to form McTNs. Cells void of McTNs also exhibited robust expression of phosphorylated cofilin (inactivated) in both an attached and detached state, indicative of a rigid cortex capable of suppressing McTN protrusion. Additional investigations are required to understand other drivers of ovarian cancer McTN formation beyond physical properties of the cell surface, including cell cycle kinetics; centrosome amplification, which has been associated with increased tumor grade, size, metastasis, and recurrence [39]; and stemness, as CD44 knockdown in glioblastoma cells has been shown to decrease McTN production [37].

Ovarian cancer ascites is rich in malignant single cells as well as cell clusters [54]. We have shown here that exposure to depolymerization agents in cells that form McTNs reduces reattachment and exposure to microtubule-stabilizing agents differentially reduces clustering relative to cells that do not form McTNs. Additional experiments are needed to understand how intra-abdominal fluid pressures, shearing forces, and temperature may influence McTN behavior. Gynecologic organs fall victim to increased abdominal pressure with ascites that may reach 22.1 mm Hg, in contrast to intra-abdominal pressures that are normally sub-atmospheric [55,56] and alterations in interstitial fluid flow as a consequence of bowel peristalsis, Valsalva, and diaphragmatic fluctuations with respiration [57]. This could have clinical relevance in the rational design of intraperitoneal drug delivery [58,59], particularly for patients who exhibit ascites, and optimizing hyperthermic intraperitoneal chemotherapy algorithms that may be linked to overall survival benefit in ovarian cancer. Currently, cisplatin is the most widely accepted agents for ovarian cancers, but a McTN-specific agent may improve the treatment of floating microscopic residual disease.

## 5. Conclusions

In summary, we have shown that floating ovarian cancer cells isolated from both cell culture and freshly harvested ascites frequently exhibit McTNs. Ovarian cancer McTN morphology and frequency of microtubule-stabilizing post-translational modifications of α-tubulin appear to be consistent with biologic metastatic potential. Because ovarian cancer McTNs are composed of tubulin, microtubule depolymerizing drugs can decrease the number and length of McTNs, while microtubule-stabilizing drugs may stabilize McTN dynamics. McTNs likely participate in key aspects of ovarian cancer metastasis and clustering, and their formation is governed in part by the physical properties of the actin cortex. McTNs represent a new and exciting therapeutic target for this disease.

## Figures and Tables

**Figure 1 cancers-14-00800-f001:**
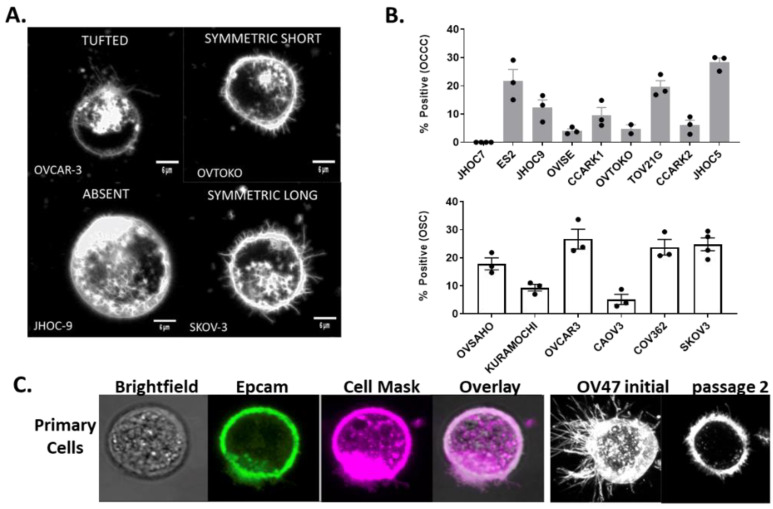
(**a**) Ovarian cancer cell lines demonstrated four distinct McTN phenotypes: absent, tufted, symmetric-long, symmetric-short. Representative images are shown. Scale shown with marker representing 6 μm (**b**). As many as 30% of ovarian serous (OSC) (white bars) and clear cell (OCCC) (grey bars) cancer cells exhibited McTNs (*n* = 10). Mean and standard error of the mean are shown. (**c**) Freshly procured non-adherent OSC cells isolated from ascites also formed McTNs, which were observed through at least two passages. (***left to right***): unstained, Epcam (green), CellMask membrane stain (magenta), and overlay. Representative cells from OV47 with McTNs from passages 0 (left) and 2 (right).

**Figure 2 cancers-14-00800-f002:**
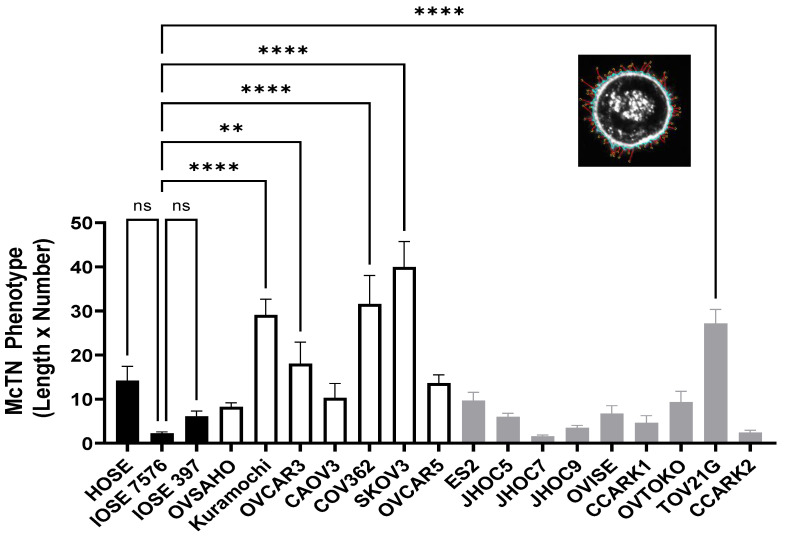
McTN phenotypes quantified by MATLAB can be described by the product of the average length of the McTNs (inset, yellow tip to blue body) by the number of McTNs per cell. Results are shown for human/immortalized ovarian surface epithelium (HOSE/IOSE) (black bars), ovarian clear cell carcinoma (OCCC) (grey bars), and ovarian serous carcinoma (OSC) (white bars) cell lines (*n* ≥1 9, 19–27, mean and standard error of the mean.) OSC and OCCC McTN length × number were compared to HOSE/IOSE McTN length × number by one-way ANOVA with a Dunnett Post Hoc Analysis with an alpha set to 0.5. ** *p* < 0.002; **** *p* < 0.0001, ns – not-significant.

**Figure 3 cancers-14-00800-f003:**
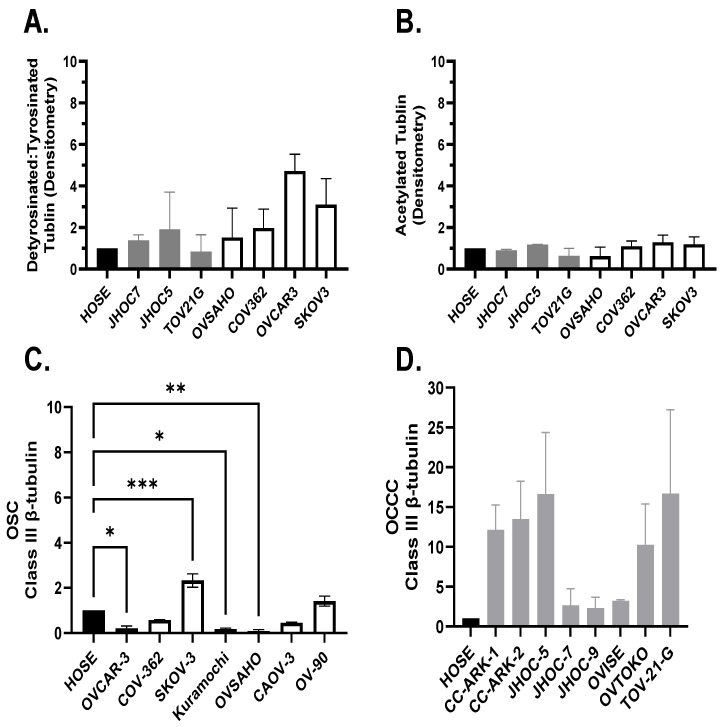
Western blot analysis of human ovarian surface epithelium (HOSE) (black bar), ovarian clear cell carcinoma (OCCC, gray bars), and ovarian serous carcinoma (OSC, white bars) cell lines. (**a**) Ovarian cancers exhibit more post-translational modifications of tubulin relative to HOSE with a trend showing that OSC cell lines (white) produce more McTNs with higher detyrosinated tubulin levels compared to OCCC (grey) (*p* = 0.29). (**b**) Acetylated tubulin expression is similar across HOSE, OSC, and OCCC (*p* = 0.61). Class III ß-tubulin is overexpressed in few OSC (1/7, white bars) (**c**) and most OCCC (5/8, grey bars) (**d**). Densitometry is shown. Data are shown in duplicate. Bars represent mean and standard error of the mean. Western blot images are provided in Appendix A. One-way ANOVA analysis with a Dunnett post hoc analysis was performed for each data set. There was a significant difference between class III ß-tubulin expression in HOSE vs. OSC cell lines OVCAR-3, SKOV-3, Kuramochi, and OVSAHO; which did not reach significance in OCCC. * *p* < 0.03; ** *p* < 0.002; *** *p* < 0.0002.

**Figure 4 cancers-14-00800-f004:**
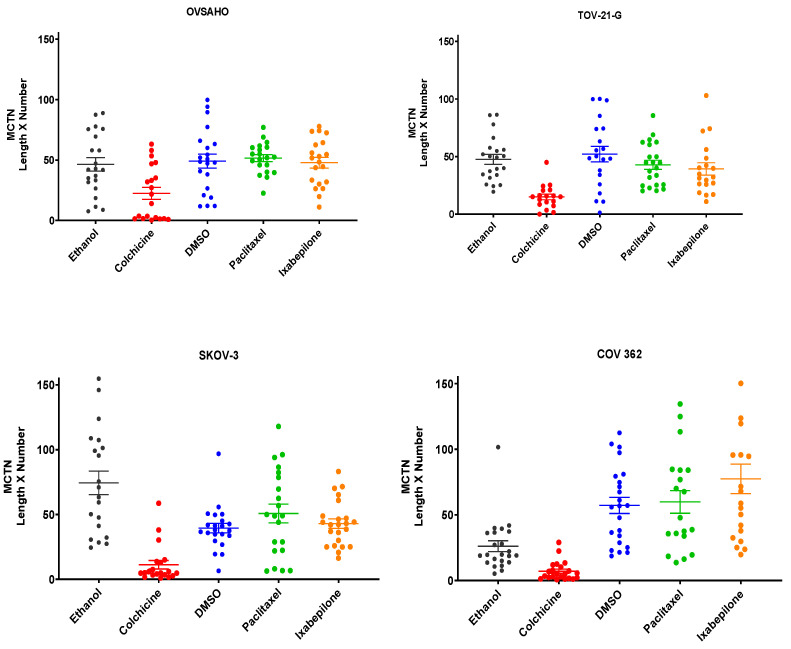
Effect of colchicine (**red**), paclitaxel (**green**), and ixabepilone (**orange**) on McTN morphology (length × number) in OSC (OVSAHO, SKOV-3, COV-362) and OCCC (TOV-21-G) lines (*n* =20). Mean and standard error of the mean are shown.

**Figure 5 cancers-14-00800-f005:**
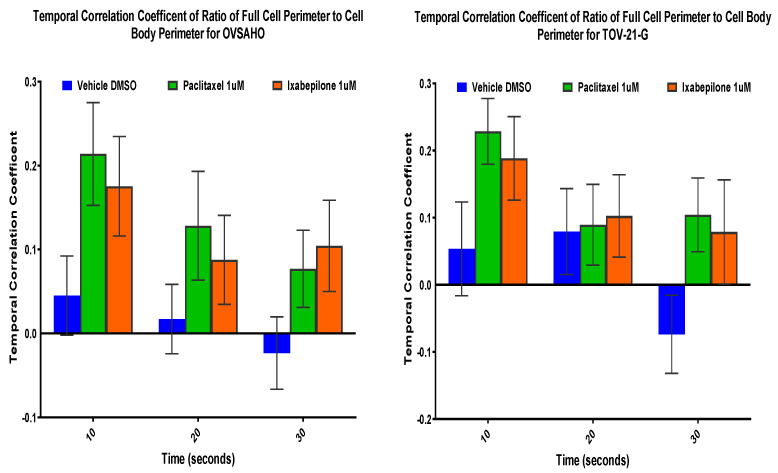
Temporal correlation coefficients following treatment of ovarian clear cell (OCCC) (i.e., TOV21G, class III ß-tubulin highexpressor) and serous (OSC) (i.e., OVASHO, class III ß-tubulin non-expressor) cell lines with paclitaxel and ixabepilone (*n* = 20). Mean with standard error of the mean are shown.

**Figure 6 cancers-14-00800-f006:**
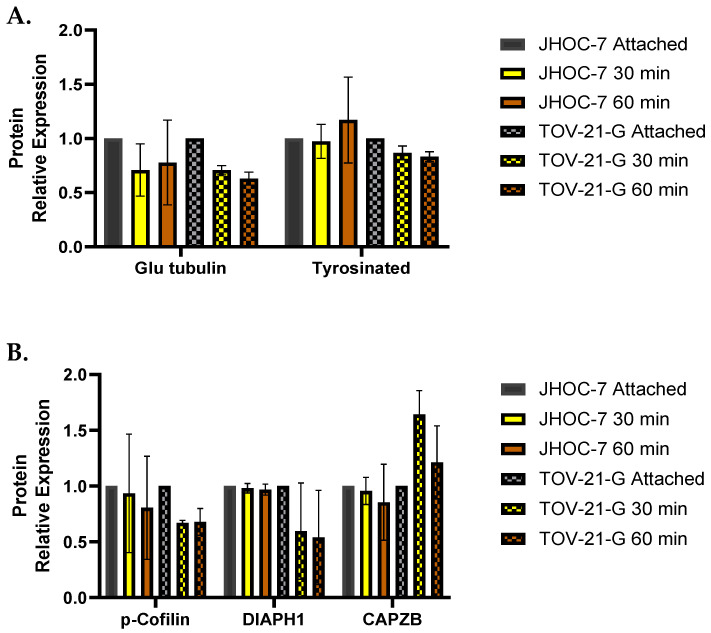
Expression of α-tubulin post-translational modifications (**a**) and actin regulatory proteins (**b**) in attached and detached cells. In the attached state, JHOC-7, the only cell line completely void of McTNs exhibited high levels of expression of tyrosinated tubulin and p-cofilin relative to McTN-forming TOV21G. Tubulin regulatory proteins were normalized to α-tubulin. Actin regulatory proteins were normalized to cofilin (*n* = 3). Mean and standard error of the mean are shown. No significant differences were found with one-way ANOVA analysis.

**Figure 7 cancers-14-00800-f007:**
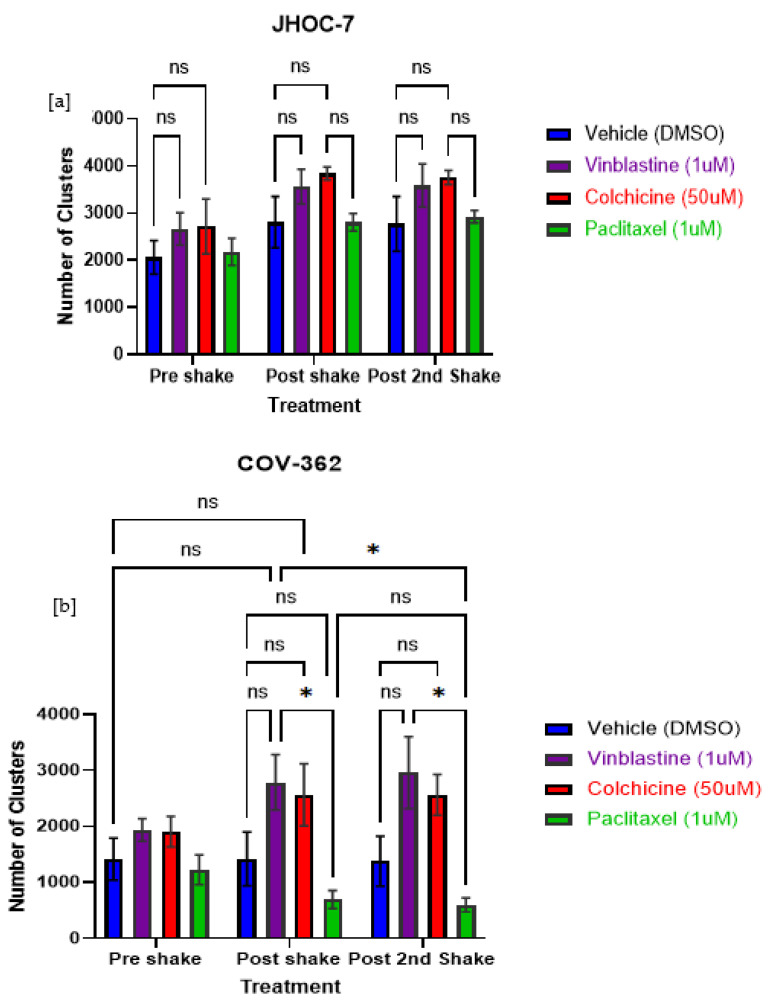
Effect of tubulin depolymerizing and stabilizing agents on ovarian cancer cell aggregation (clusters). Cells were incubated for 18–20 h with drug. The number of clusters formed after two 1-min agitations is shown (shake). OSC COV-362, which demonstrates predominantly as a symmetric long phenotype, demonstrated reduced clustering ability (i.e., fragmentation into more numerous aggregates) after treatment with a microtubule depolymerizing agent (vinblastine, cochicine) compared to vehicle. OCCC JHOC7, which displays an entirely absent McTN phenotype, demonstrated fewer differences between different drug treatments. This alludes to a role for McTNs in ovarian cancer cell clustering. Mean and standard deviation are shown. Data were analyzed using a two-way ANOVA with Tukey’s multiple comparison test with a family-wise alpha threshold and confidence level set to 0.05. (**a**) For JHOC7, a two-way ANOVA was performed to analyze the effect of drug treatment and aggregate disruption (shaking) on the number of cell clusters. A two-way ANOVA revealed that there was no statistically significant interaction between the effects of drug treatments and shaking (F(6,24) = 0.08849, *p* = 0.997). Simple main effects analysis showed that agitations (shaking) did have a statistically significant effect on cellular aggregation (*p* = 0.0049). Simple main effects analysis showed that the drug treatment did have a statistically significant effect on clusters (*p* = 0.0167). Multiple comparisons post hoc analysis demonstrated no significant differences between drug treatments and agitations (shaking). (**b**) For COV-362, a two-way ANOVA revealed that there was not a statistically significant interaction between the effects of drug treatments and shaking (F(6,24) = 0.9963, *p* = 0.4503). Simple main effects analysis showed that shaking did not have a statistically significant effect on clusters (*p* = 0.5965). Simple main effects analysis showed that the drug treatment did have a statistically significant effect cluster number (*p* < 0.0001,*). Each condition was performed in triplicate. The experiment was performed in triplicate. ns-notsignificant

**Table 1 cancers-14-00800-t001:** Morphologies observed across ovarian clear cell (OCCC) (grey bars) and serous (OSC) (white bars) cell lines.

	OCCC								OSC						
	CCARK1	CCARK2	OVTOKO	OVISE	JHOC5	JHOC7	JHOC9	ES2	TOV21G	Kuramochi	OVSAHO	CAOV3	COV362	SKOV3	OVCAR3
**Observations (n)**	25	22	22	27	24	26	24	24	26	23	26	25	19	26	25
**A (%)**	64	82	23	48	25	100	67	13	0	0	23	16	21	4	38
**SS (%)**	28	18	64	26	75	0	25	42	46	52	77	60	32	62	13
**SL (%)**	0	0	14	0	0	0	0	46	54	48	0	28	47	35	13
**T (%)**	8	0	0	26	0	0	8	0	0	0	0	4	0	0	38
**Dominant** **Morphology**	A	A	SS	A	SS	A	A	SL/SS	SL	SS	SS	SS	SL	SS	T/A

Observations (n): The number of cells analyzed is provided in the first row. Microtentacle morphology: A—absent, SS—symmetrical short, SL—symmetrical long, T—tufted.

**Table 2 cancers-14-00800-t002:** Half maximal inhibitory concentrations for paclitaxel and ixabepilone in cell lines used in this study.

Treatment	CC-ARK-1	CC-ARK-2	OVTOKO	JHOC-5	JHOC-7	TOV-21-G	ES-2	Kuramochi	OVSAHO	CaOV3	COV-362	SKOV-3
**Paclitaxel**	0.788	0.436	4.32	0.368	1.72	0.35	1.51	1.51	0.276	2.77	2.72	1.41
**Ixabepilone**	2.48	2.01	2.64	1	0.965	0.83	2.99	3.81	5.37	0.315	0.315	0.6

## Data Availability

The data presented in this study are available on request from the corresponding author. The data are not publicly available due to limitations- on patient data under the IRB protocol GCC1488.

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
