# Peer review of "Microtentacle Formation in Ovarian Carcinoma"

_cancers, 2022, doi:10.3390/cancers14030800_

Round 1
Reviewer 1 Report
The manuscript submitted by Reader et al. reports for the first time the evidence of microtentacles (McTNs) in ovarian cancer, highlighting the novelty of the findings. The topic is interesting and deserves elucidations.
The authors well describe the state-of-art of McTNs biology, even though they can extend their discussion focusing on other possible therapeutic strategies to target McTNs in addition to microtubule-stabilizing and depolymerazing agents (for example compounds/drugs that modulate post-transcriptional modification of cytoskeletal elements).
Additionally, a more detailed description of the methodologies used in the study could be helpful for the readers that are not experts in the field of McTNs biology.
The discussion/conclusions could be improved by adding the authors' opinions/perspectives concerning (i) the translational relevance of targeting McTNs and (ii) the possible role of McTNs in ovarian normal surface epithelial cells.
Overall the paper is quite well written.
Author Response
Please see the attachment, which includes the untracked revised manuscript for clarity of referenced text by line number, as well as low-resolution versions of the Supplemental Figures. The response to the Reviewer follows at the end.

Reviewer 2 Report
This work is relevant and concerns the problem of identifying additional mechanisms of metastasis of ovarian tumors.
However, major points must be addressed.
The authors have to compare detached floating tumor cells isolated from the ascitic fluid of patients with different stages (not only stage IV, but also stages II-III) on the presence of McTN. It is also necessary to compare the presence of McTN on tumor cells in the ascitic fluid of patients before and after paclitaxel-based chemotherapeutic treatment (or patients with or without chemotherapy). The analysis of the expression of EMT markers and metastasis-associated markers on tumor cells having different forms of McTN would increase the quality of manuscript. Invasion assay can help to understand how tumor cells with particular charachteristics can differ.
Did cell lines differ by stages? Were there any cell lines with stage IV?
It is not entirely clear how paclitaxel affected the amount and type of McTN. A general conclusion is needed as this is a clinically important issue.
All figures require correstions. They must be diminished and combined. The quality and colors of figures are poor - need to be corrected. There are no standard deviations and no p-value in some Figures.
Author Response

(The authors gave the same response as above.)

Reviewer 3 Report
This study shows the profiling of microtentacles in ovarian cancer cell lines.
Microtentacles are mainly reported in the breast cancer study.
Their study is the first study of profiling in ovarian cancer.
The quality of presented data is enough to consider the publication. Figure 6 is not able to see it is because the color is dark. The color should be changed.
And in western blotting analysis, the authors should show the raw data of the membrane as supplementary figures.
Author Response

(The authors gave the same response as above.)
